

# Finding melanoma drugs through a probabilistic knowledge graph

Jamie Patricia McCusker[1], Michel Dumontier[2], Rui Yan[1], Sylvia He[1], Jonathan S. Dordick[3,4] and Deborah L. McGuinness[1,4]

[1] Department of Computer Science, Rensselaer Polytechnic Institute, Troy, NY, USA
[2] Stanford Center for Biomedical Informatics Research, Stanford University School of Medicine, Stanford, CA, USA
[3] Department of Chemical & Biological Engineering, Rensselaer Polytechnic Institute, Troy, NY, USA
[4] Center for Biotechnology & Interdisciplinary Studies, Rensselaer Polytechnic Institute, Troy, NY, USA

## ABSTRACT

Metastatic cutaneous melanoma is an aggressive skin cancer with some progression-slowing treatments but no known cure. The omics data explosion has created many possible drug candidates; however, filtering criteria remain challenging, and systems biology approaches have become fragmented with many disconnected databases. Using drug, protein and disease interactions, we built an evidence-weighted knowledge graph of integrated interactions. Our knowledge graph-based system, ReDrugS, can be used via an application programming interface or web interface, and has generated 25 high-quality melanoma drug candidates. We show that probabilistic analysis of systems biology graphs increases drug candidate quality compared to non-probabilistic methods. Four of the 25 candidates are novel therapies, three of which have been tested with other cancers. All other candidates have current or completed clinical trials, or have been studied in in vivo or in vitro. This approach can be used to identify candidate therapies for use in research or personalized medicine.

## INTRODUCTION

Metastatic cutaneous melanoma is an aggressive cancer of the skin with low prevalence but very high mortality rate, with an estimated 5-year survival rate of 6% (*Barth, Wanek & Morton, 1995*). There are currently no known therapies that can consistently cure metastatic melanoma. Vemurafenib is effective against BRAF mutant melanomas (*Chapman et al., 2011*) but resistant cells often result in recurrence of metastases (*Le et al., 2013*). Melanoma itself may be best approached based on the individual genetics of the tumor, as it has been shown to involve mutations in many different genes to produce the same disease (*Krauthammer et al., 2015*). Because of this, an individualized approach may be necessary to find effective treatments.

Drug repurposing, or the discovery of new uses for existing approved drugs, can often lead to effective new treatments for diseases. A wide range of computational methods have been developed in support of drug repositioning. Computational approaches (*Sanseau & Koehler, 2011*) include topic modeling (*Bisgin et al., 2012, 2014*), side-effect

Corresponding authors
Jamie Patricia McCusker,
mccusj@cs.rpi.edu
Deborah L. McGuinness,
dlm@cs.rpi.edu

similarity (*Yang & Agarwal, 2011*; *Ye, Liu & Wei, 2014*), drug and/or disease similarity (*Chiang & Butte, 2009*; *Gottlieb et al., 2011*), genome-wide association studies (*Kingsmore et al., 2008*; *Grover et al., 2014*), and gene expression (*Lamb et al., 2006*; *Sirota et al., 2011*). Systems biology has also provided a number of network analysis approaches (*Yang & Agarwal, 2011*; *Wu, Wang & Chen, 2013*; *Cheng et al., 2012*; *Emig et al., 2013*; *Harrold, Ramanathan & Mager, 2013*; *Wu et al., 2013*; *Vogt, Prinz & Campillos, 2014*) but the field has been limited by a fragmentation of databases. Most systems biology databases are not aligned with each other, and typically leave out crucial information about how other biological entities, like drugs and diseases, interact with the systems biology graph. Further, while some interaction databases provide human curation and validation of pathway interactions, and others provide experimental evidence for the recorded interactions, there has not yet been, to our knowledge, a resource that combines the two approaches and quantifies the reliability of the evidence used to assert the interactions.

A knowledge graph is a compilation of facts and figures that can be used to provide contextual meaning to searches. Google is using knowledge graphs to improve its search and to analyze the information graph of the web; Facebook is using them to analyze the social graph. We built our knowledge graph with the goal of unifying large parts of biomedical domain knowledge for both mining and interactive exploration related to drugs, diseases, and proteins. Our knowledge graph is enhanced by the provenance of each fragment of knowledge captured, which is used to compute the confidence probabilities for each of those fragments. Further, we use open standards from the world wide web consortium (W3C), including the resource description framework (RDF) (*Richard, David & Markus, 2014*), web ontology language (OWL) (*Motik, Patel-Schneider & Cuenca Grau, 2009*), and SPARQL (*Harris, Seaborne & Prud'hommeaux, 2013*). The representation of the knowledge in our knowledge graph is aligned with best practice vocabularies and ontologies from the W3C and the biomedical community, including the provenance ontology (PROV-O) (*Lebo, Sahoo & McGuinness, 2013*), the HUPO proteomics standards initiative molecular interactions (PSI-MI) ontology (*Hermjakob et al., 2004*), and the semanticscience integrated ontology (SIO) (*Dumontier et al., 2014*). Use of these standards, vocabularies, and ontologies make it simple for ReDrugS to integrate with other similar efforts in the future with minimal effort.

We proposed and built a novel computational drug repositioning platform, that we refer to as ReDrugS, that applies probabilistic filtering over individually-supported assertions drawn from multiple databases pertaining to systems biology, pharmacology, disease association, and gene expression data. We use our platform to identify novel and known drugs for melanoma.

## RESULTS

We used ReDrugS to examine the drug–target–disease network and identify known, novel, and well supported melanoma drugs. The ReDrugS knowledge base contained 6,180 drugs, 3,820 diseases, 69,279 proteins, and 899,198 interactions. The drugs included in ReDrugS follow the distribution by the Anatomic Therapeutic Classification (ATC) categories shown in Fig. 1.

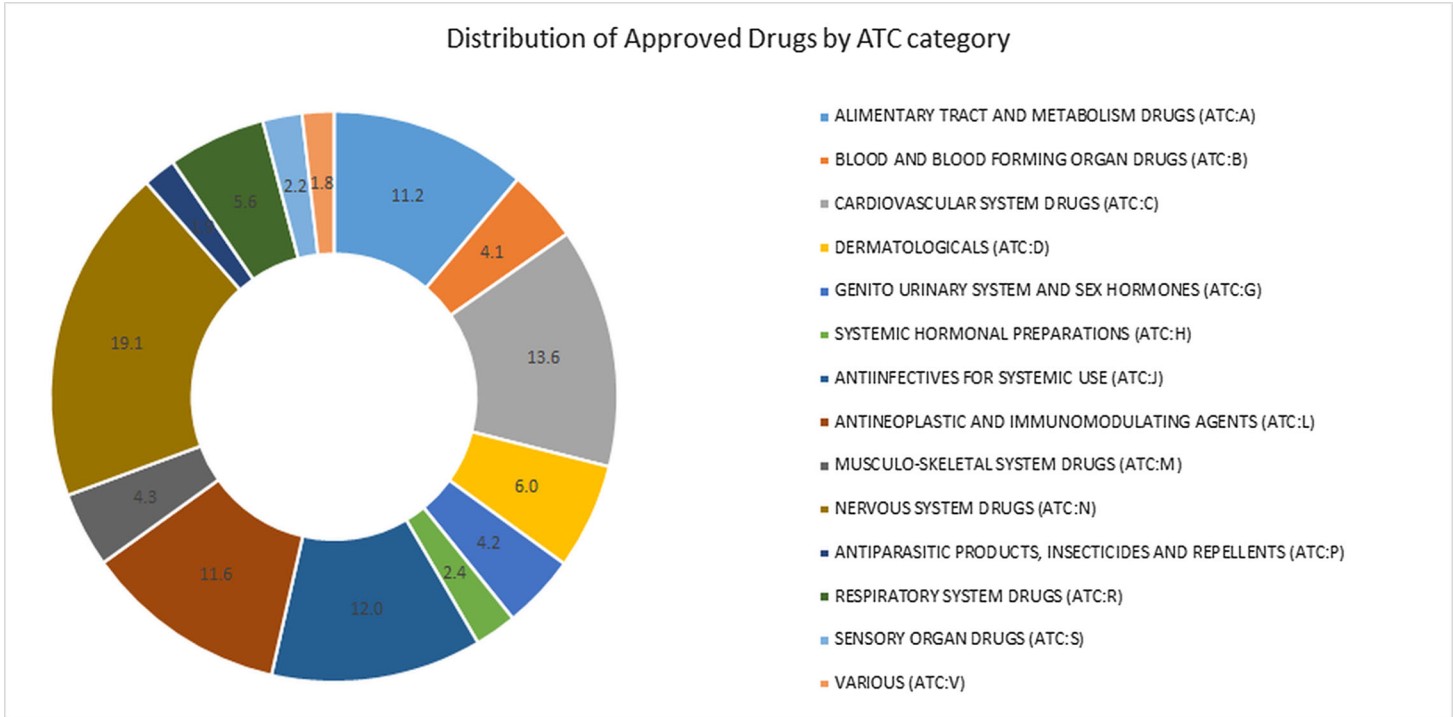

**Figure 1 Percentage approved drugs in each of the categories of the anatomic therapeutic classification (ATC) system.**

We examined drug and gene connections that were three or less interaction steps from melanoma, and additionally filtered interactions with a joint probability greater or equal to 0.93. We identified 25 drugs in the resulting drug–gene–disease network surrounding melanoma as illustrated in Fig. 2.

We then validated the set of 25 drugs by determining their position in the drug discovery pipeline for melanoma. Table 1 shows that nearly all drugs uncovered by ReDrugS were previously been identified as potential melanoma therapies either in clinical trials or in vivo or in vitro. Of the 25 drugs, 12 have been in Phase I, II, or III clinical trials, five have been studied in vitro, four in vivo, one was investigated as a case study, and three are novel.

To further evaluate our system, we examined the impact of decreasing the joint probability or increasing the number of interaction steps. Figures 3A and 3B show precision, recall, and f-measure curves while varying each parameter. Using these information retrieval performance curves, we found that using a joint probability of 0.93 or greater with three or less interaction steps maximizes the precision and recall as shown in Fig. 3.

By performing a sampled literature search on hypothesis candidates with a joint probability of 0.5 or higher and six or fewer interaction steps, we were able to generate precision, recall, and f-measure curves for both cutoffs to find our cutoff of 0.93 with three or fewer interaction steps. The precision, recall, and f-measure curves are shown for varying joint probability thresholds in Fig. 3A and for varying interaction step counts in Fig. 3B.

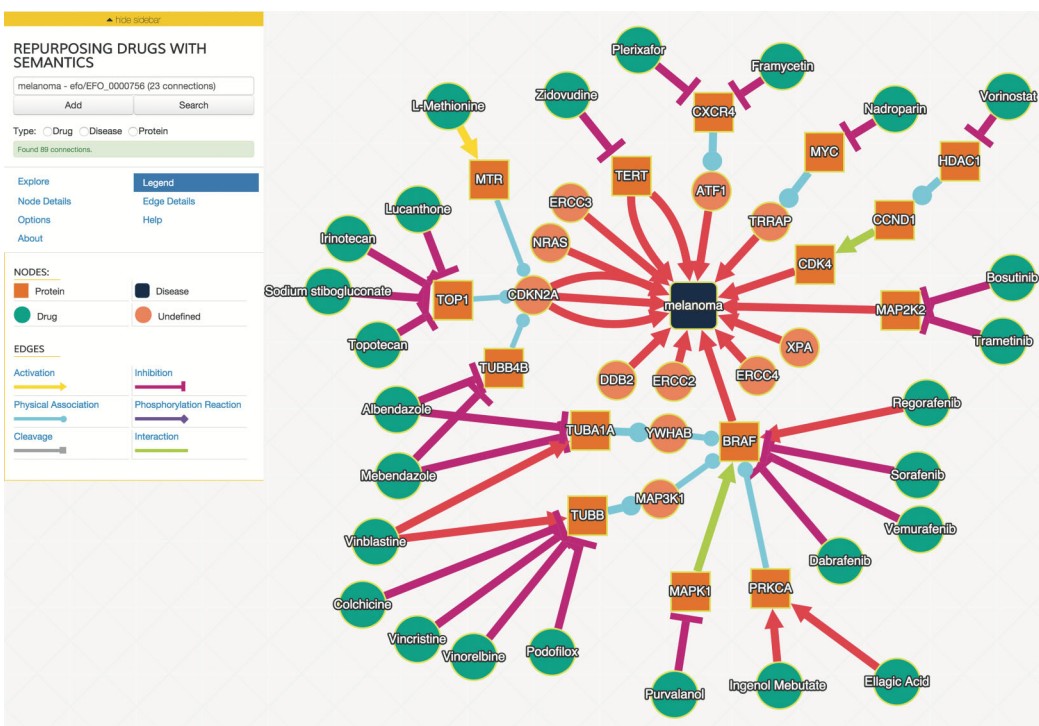

**Figure 2** **The interaction graph of predicted melanoma drugs with a probability of 0.93 or higher and have three or fewer intervening interactions between drug and disease.** The "Explore" tab contains the controls to expand the network in various ways, including the filtering parameters. Node and edge detail tabs provide additional information about the selected node or edge, including the probabilities of the edges selected. Users can control the layout algorithm and related options using the "Options" tab.

# DISCUSSION

We designed ReDrugS to quickly and automatically integrate and filter a heterogeneous biomedical knowledge graph to generate high-confidence drug repositioning candidates. Our results indicate that ReDrugs generates clinically plausible drug candidates, in which half are in various stages of clinical trials, while others are novel or are being investigated in pre-clinical studies. By helping to consolidate the three main datatypes—drug targets, protein interactions, and disease genes—ReDrugs can amplify the ability of researchers to filter the vast amount of information into those that are relevant for drug discovery.

## Candidate significance

Three drugs were identified that have not previously been studied for melanoma treatment. Framycetin, a CXCR4 inhibitor, has not previously been considered for melanoma treatment. While it is nephrotoxic when administered orally (*Greenberg, 1965*), it is used topically as an antibacterial treatment. While it may not be of use for metastasis, it might serve as a simple, inexpensive prophylactic treatment after excision of primary tumors. Additionally, Lucanthone and Podofilox were identified as having potential effects on melanoma through CDKN2A and MAP kinase, respectively.

**Table 1 Drug discovery status for 25 drug candidates identified using ReDrugS.**

| Status | Drug | Pathway | Steps | Joint $p$ |
|---|---|---|---|---|
| Approved | Vemurafenib (*Chapman et al., 2011*) | BRAF | 2 | 0.98 |
| Phase III | Dabrafenib (*Hauschild et al., 2012*) | BRAF | 2 | 0.98 |
| | Sorafenib (*National Cancer Institute, 2005*) | BRAF | 2 | 0.98 |
| | Vinblastine (*Luikart, Kennealey & Kirkwood, 1984*) | MAP kinase | 3 | 0.93 |
| Phase II | Zidovudine (*Humer et al., 2008*) | TERT | 2 | 0.98 |
| | Trametinib (*Kim et al., 2012*) | MAP kinase | 2 | 0.98 |
| | Regorafenib (*Istituto Clinico Humanitas, 2015*) | BRAF | 2 | 0.98 |
| | Nadroparin (*Nagy, Turcsik & Blaskó, 2009*) | MYC | 3 | 0.97 |
| | Vinorelbine (*Whitehead et al., 2004*) | MAP kinase | 3 | 0.93 |
| | Irinotecan (*Fiorentini et al., 2009*) | CDKN2A | 3 | 0.93 |
| | Topotecan (*Kraut et al., 1997*) | CDKN2A | 3 | 0.93 |
| Phase I | Sodium stibogluconate (*Naing, 2011*) | CDKN2A | 3 | 0.93 |
| Case study | Ingenol mebutate (*Mansuy et al., 2014*) | PRKCA/BRAF | 3 | 0.95 |
| In vitro | Bosutinib (*Homsi et al., 2009*) | MAP kinase | 2 | 0.98 |
| | Purvalanol (*Smalley et al., 2007*) | MAP kinase/TP53 | 3 | 0.97 |
| | Ellagic acid (*Kim et al., 2008*) | PRKCA/BRAF | 3 | 0.95 |
| | Albendazole (*Patel et al., 2011*) | CDKN2A | 3 | 0.93 |
| | Colchicine (*Lemontt, Azzaria & Gros, 1988*) | MAP kinase | 3 | 0.93 |
| In vivo | Plerixafor (*D'Alterio et al., 2012*) | CXCR4 | 3 | 0.97 |
| | Vincristine (*Sawada et al., 2004*) | MAP kinase | 3 | 0.93 |
| | L-Methionine (*Clavo & Wahl, 1996*) | CDKN2A | 3 | 0.93 |
| | Mebendazole (*Doudican et al., 2008*) | CDKN2A | 3 | 0.93 |
| Novel | Framycetin | CXCR4 | 3 | 0.97 |
| | Lucanthone | CDKN2A | 3 | 0.93 |
| | Podofilox | MAP kinase | 3 | 0.93 |

**Note:**
"Pathway" refers to the target or pathway that the drug acts on. "Steps" is distance in number of interactions between the drug and the disease, and "Joint p" is the joint probability that all of those interactions occur.

One drug we identified, Vemurafenib, is approved for treatment of late stage melanoma has been shown to inhibit the BRAF protein in BRAF-V600 mutant melanomas (*Chapman et al., 2011*). However, cells can become resistant to Vemurafenib, thereby leading to metastasis (*Le et al., 2013*).

A number of the drugs we identified are in clinical trials for treatment of melanoma. We identified BRAF-oriented drugs, Dabrafenib (*Hauschild et al., 2012*), Sorafenib (*National Cancer Institute, 2005*), and Regorafenib (*Istituto Clinico Humanitas, 2015*), that have been evaluated in clinical trials, but have not yet been approved. Zidovudine or Azidothymidine is a TERT inhibitor that has shown significant melanoma tumor reductions in mouse models (*Humer et al., 2008*). Three MAP kinase-related compounds, Vinblastine (*Luikart, Kennealey & Kirkwood, 1984*), Trametinib (*Kim et al., 2012*), and Vinorelbine (*Whitehead et al., 2004*) were identified that are in clinical trials for melanoma treatment. CDKN2A was another popular target, as Irinotecan (*Fiorentini et al., 2009*),

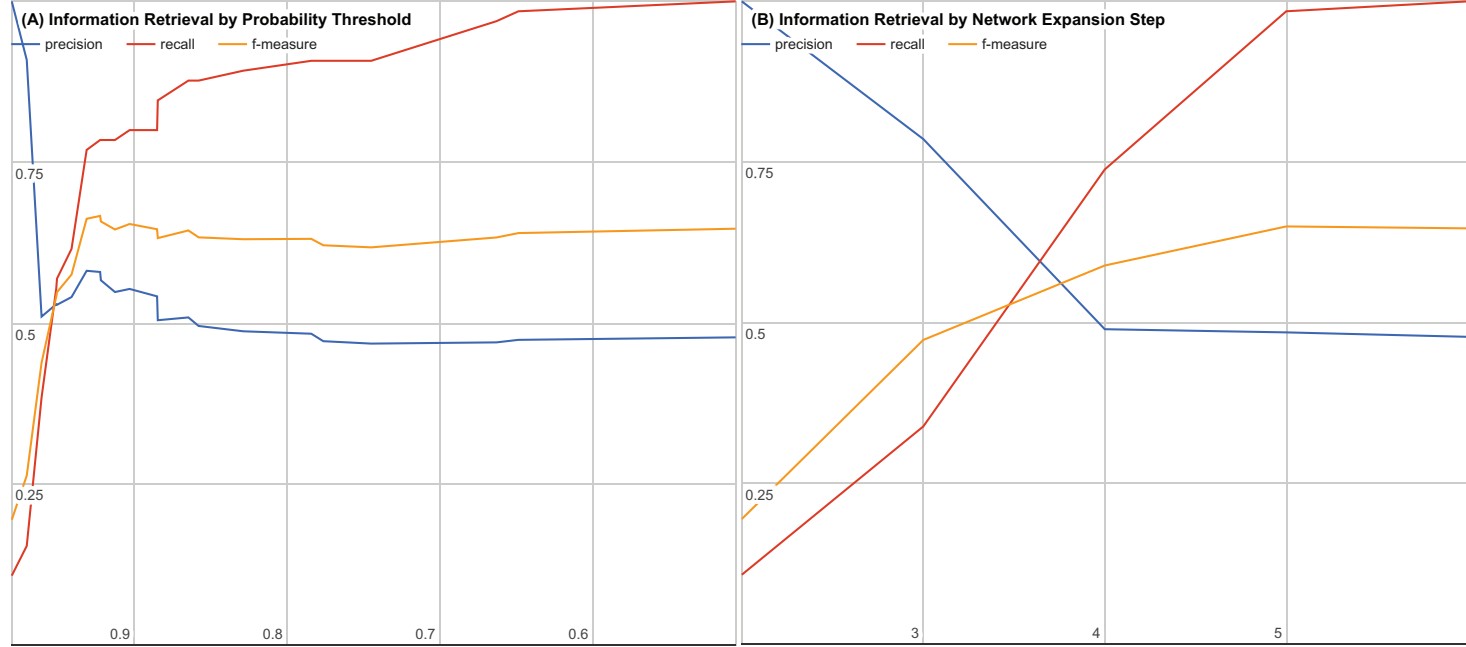

**Figure 3  Precision, recall, and f-measure by (A) varying thresholds for joint probability and (B) varying number of interaction steps.** Precision is the percentage of returned candidates that have been validated experimentally or have been in a clinical trial (a "hit") versus all candidates returned. Recall is the percentage of all known validated "hits." f-Measure is the geometric mean of precision and recall that provides a balanced evaluation of the quality and completeness of the results.

Topotecan (*Kraut et al., 1997*), and Sodium stibogluconate (*Naing, 2011*) are all drugs in clinical trial that we identified as potential therapies.

Many other drugs were identified that are being studied in the lab. Additional drugs were identified that target the MAP kinase pathway, including Bosutinib (*Homsi et al., 2009*), Purvalanol (*Smalley et al., 2007*), Colchicine (*Lemontt, Azzaria & Gros, 1988*), and Vincristine (*Sawada et al., 2004*). Podofilox has not yet been investigated in melanoma treatments, but preliminary investigations have focused on treating chronic lymphocytic leukemia (*Shen et al., 2013*) and non-small cell lung cancer (*Peng et al., 2014*). Since these drugs attack MAPK2 and related proteins rather than BRAF or NRAS, they can potentially synergize with other treatments (*Homsi et al., 2009*). Bosutinib in particular has been investigated as a synergistic treatment for melanoma (*Held et al., 2012*). Another possible treatment pathway is CXCR4 inhibition. Mouse models suggest that CXCR4 inhibitors like Plerixafor can reduce tumor metastasis and primary tumor growth (*D'Alterio et al., 2012*). We identify both Plerixafor and Framycetin (Neomycin B) as useful CXCR4 inhibitors. Two PKRCA activators, Ingenol mebutate and Ellagic acid, were also identified. PKRCA binds with BRAF (*Pardo et al., 2006*), but it is mechanistically unclear how PKRCA activation would result in treatment of melanoma. A number of other therapies are also notable. Purvalenol can inhibit GSK3$\beta$, which in turn activates TP53. Some, but not all, melanomas have TP53 deactivation (*Smalley et al., 2007*). Nadroparin, a MYC inhibitor, may inhibit tumor progression (*Nagy, Turcsik & Blaskó, 2009*). More broadly, heparins can potentially inhibit the metastatic process in melanoma and other cancers (*Maraveyas et al., 2010*).

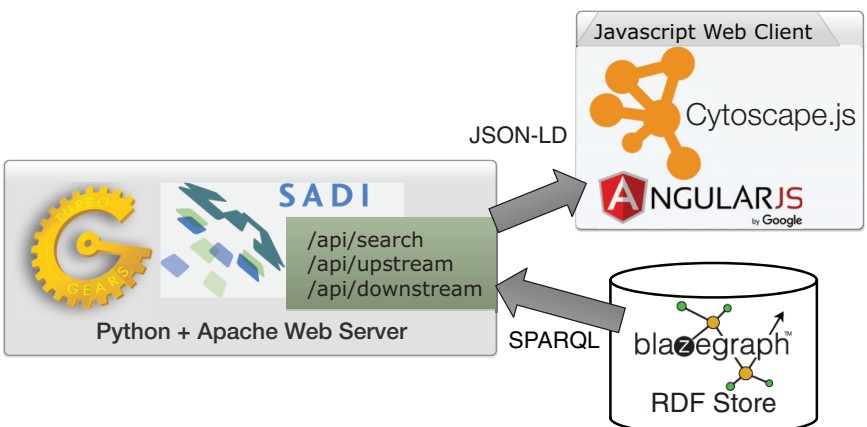

**Figure 4 The ReDrugS software architecture.** Using web standards and a three-layer architecture (RDF store, web server, and rich web client), we were able to build a complete knowledge graph analysis platform.

The approach that we present here offers a novel, mechanism-focused exploration to identify and examine drugs and targets related to cancer. This approach filters our noisy or poorly supported parts of the knowledge graph to identify more confident mechanisms between drugs, targets, and diseases. Thus, our approach can be used to explore high confidence associations that are produced as a result of large scale computational screens that use network connectivity (*Yang & Agarwal, 2011*; *Wu, Wang & Chen, 2013*; *Cheng et al., 2012*; *Emig et al., 2013*; *Harrold, Ramanathan & Mager, 2013*; *Wu et al., 2013*; *Vogt, Prinz & Campillos, 2014*), the complementarity in drug-disease gene expression, and the similarity of chemical fingerprints, side-effects, targets, or indications (*Yang & Agarwal, 2011*; *Ye, Liu & Wei, 2014*; *Chiang & Butte, 2009*; *Gottlieb et al., 2011*; *Lamb et al., 2006*; *Sirota et al., 2011*). Importantly, since we focus on protein networks that are strongly linked with diseases, we believe that our mechanism focused approach will also aid in the identification of disease-modifying drug candidates, rather than solely those that would be useful for the treatment of symptomatic phenotypes or related co-morbid conditions.

## Architecture

ReDrugS uses a fairly straightforward web architecture, as shown in Fig. 4. It uses the Blazegraph RDF database backend. The database layer is interchangeable except that the full text search service needs to use Blazegraph-only properties to perform text searches as text indexing is not yet standardized in the SPARQL query language. All other aspects are standardized and should work with other RDF databases without modification. ReDrugs currently uses the Python-based TurboGears web application framework hosted using the web services gateway interface standard via an Apache HTTP server. TurboGears in turn hosts the semantic automated discovery and integration (SADI) web services that drive the application and access the database. It also serves up the static HTML and supporting files.

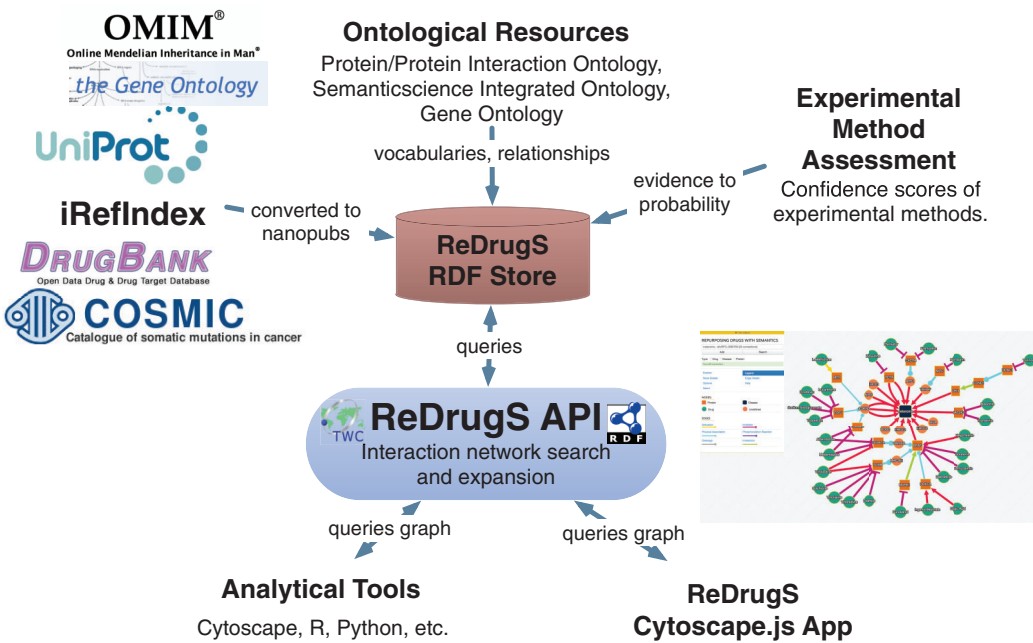

**Figure 5 The ReDrugS data flow.** Data is selected from external databases and converted using scripts into nanopublication graphs, which are loaded into the ReDrugS data store. This is combined with experimental method assessments, expressed in OWL, and public ontologies into the RDF store. The web service layer queries the store and produces aggregate analyses of those nanopublications, which is consumed and displayed by the rich web client. The same APIs can be used by other tools for further analysis.

The user interface is implemented with AngularJS and Cytoscape.js, which submits queries to the SADI web services using JSON-LD and aggregates results into the networked view. The software relies exclusively on standardized protocols (HTTP, SADI, SPARQL, RDF, and others) to make it simple to replace technologies as needed. The data itself is processed using conversion scripts as shown in Fig. 5.

We have also adapted and featured ReDrugS in an immersive visualization laboratory called the collaborative-research augmented immersive virtual environment (CRAIVE) lab at RPI, as shown in Fig. 6. The goal of the demonstration was to explore new ways to visualize, sonify, and interact with big data in large-scale virtual reality systems. We also leveraged a gesture controller (Microsoft kinect) to interact with the visualization. With the 360° projection, multiple people can explore the visualization concurrently, which accelerates the exploration and discovery speed.

## Limitations and future work

Our study has a some limitations. First, our study is limited by the sources of data used. We used three databases (DrugBank, iRefIndex, and online Mendelian inheritance in man (OMIM)) to construct the initial knowledge graph. These databases are continuously changing and necessarily incomplete with respect to the total number of drugs, targets, protein interactions, diseases, and disease genes. For instance, as of 8/15/2016 there are over 2,000 additional FDA approved drugs in DrugBank than in the version that was initially used. Second, the focus of our work is on the potential repositioning of FDA

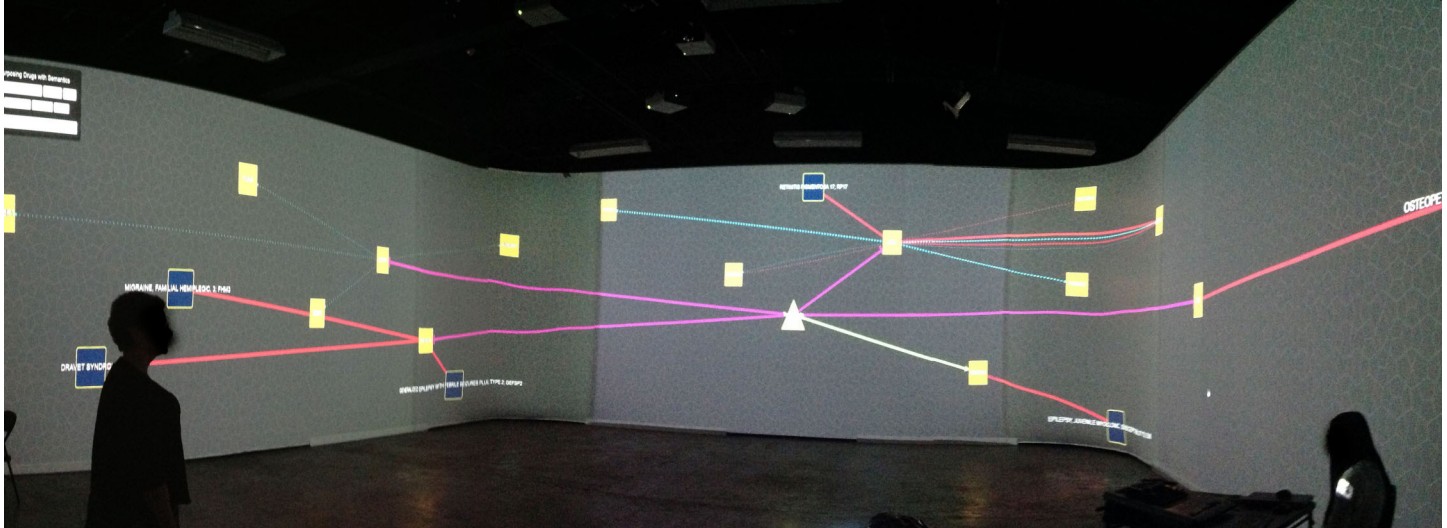

**Figure 6** The authors demonstrate the ReDrugS user interface in the collaborative-research augmented immersive virtual environment (CRAIVE) lab at RPI.

approved drugs, which means that tens of thousands of chemical compounds with protein binding activity cannot be considered as candidates in the current study. Third, our path expansion is currently limited to pairwise protein–protein interactions, which excludes interactions as a result of protein complexes or regulatory pathways. Having a more sophisticated understanding of non-direct interactions will help identify candidate drugs that can regulate entire pathways in a more rational manner. Additionally, we aim to incorporate knowledge of the complementarity of drug and disease gene expression patterns as evidenced by the connectivity map (*Lamb et al., 2006*), which could suggest therapeutic and adverse interactions. Finally, as we develop new hypotheses about potential new drug effects, we plan to test them using a new three-dimensional cellular microarray to perform high-throughput drug screening (*Lee et al., 2008*) with reference samples. The integration of computational predictions and high-throughput screening platform will enable the systematic evaluation of any drug or mechanism of action against any disease or adverse event.

## MATERIALS AND METHODS

This research project did not involve human subjects. The ReDrugS platform consists of a graphical web application, an application programming interface (API), and a knowledge base. The graphical web application enables users to initiate a search using drug, gene, and disease names and synonyms. Users can then interact with the application to expand the network at an arbitrary number of interactions away from the entity of interest, and to filter the network based on a joint probability between the source and target entities. Drug–protein, protein–protein, and gene–disease interactions were obtained from several datasets and integrated into ontology-annotated and provenance and evidence bearing representations called nanopublications. The web application obtains information from the knowledge base using semantic web services. Finally, we evaluated our approach by

examining the mechanistic plausibility of the drug in having melanoma-specific disease modifying ability. We evaluated a large number of possible drug/disease associations with varying joint probabilities and interaction steps to determine the thresholds with the highest f-measure, resulting in our thresholds of three or less interactions and a joint probability of 0.93 or higher.

Using the ReDrugS application page (http://redrugs.tw.rpi.edu) we initiate our search for "melanoma," and select the first suggestion obtained from the experimental factor ontology (EFO) (http://www.ebi.ac.uk/efo/EFO_0000756). The application then provides immediate neighborhood of drugs and genes that are associated with melanoma. We expanded the network by first selecting the melanoma node and expanding the link distance to $|l| \leq 3$ and changing the minimum joint probability to $p \geq 0.93$ in the search options. Importantly, we also limit the node type to "Drug." Finally, we click on the "find incoming links" button (two left-facing arrows). When finished the network will show all drugs interacting with melanoma that meet the above criteria, as well as any intervening entities and their interactions. The resulting network can be downloaded as an image, or a summary CSV file. We used the CSV file to validate the links by searching Google Scholar and ClinicalTrials.gov for each proposed drug/disease combination. We consider a "hit" to be a pairing with a published positive experiment in vivo or in vitro or any pairing that has been tested in a clinical trial. While this level of validation does not guarantee efficacy, it does determine if the resulting connection is a plausible hypothesis that might be tested.

## Data fusion

We developed a structured knowledge base containing data pertaining to drugs, targets, interactions, and diseases. We used five data sources: iRefIndex (*Razick, Magklaras & Donaldson, 2008*), DrugBank (*Wishart et al., 2006*), UniProt gene ontology annotations (GOA) (*Camon et al., 2004*), the online Mendelian inheritance in man (OMIM) (*Hamosh et al., 2005*), and the catalogue of somatic mutations in cancer (COSMIC) gene census (*Futreal et al., 2004*).

iRefIndex contains protein–protein interactions and protein complexes and is an amalgam of the biomolecular interaction network database (*Bader, Betel & Hogue, 2003*), BioGRID (*Stark et al., 2006*), the comprehensive resource of mammalian protein complexes (*Ruepp et al., 2010*), database of interacting proteins (*Xenarios et al., 2002*), human protein reference database (*Keshava Prasad et al., 2009*), InnateDB (*Lynn et al., 2008*), IntAct (*Kerrien et al., 2011*), MatrixDB (*Chautard et al., 2011*), molecular interaction database (*Chatr-aryamontri et al., 2008*), MPact (*Güldener et al., 2006*), microbial protein interaction database (*Goll et al., 2008*), MIPS mammalian protein–protein interaction database (*Pagel et al., 2005*), and online predicted human interaction database (*Brown & Jurisica, 2005*). DrugBank provides information about experimental/approved drugs and their targets, and UniProt GOA describes proteins in terms of their   biological processes, cellular locations, and molecular functions. OMIM provides associations between genes and inherited or genetically-driven diseases. The COSMIC gene census is a curated list of genes that have causal associations with one or more cancer types.

Each association (e.g., drug–target, protein–protein, disease–gene) was captured using the nanopublication (*Groth, Gibson & Velterop, 2010*) scheme. A nanopublication is a digital artifact that consists of an assertion, its provenance, and information about the digital publication. Our nanopublications are represented as linked data: each data item is identified using an dereferenceable HTTP uniform resource identifier (URI) and statements are represented using the RDF. Each nanopublication corresponds to a single interaction assertion from one of the databases. We used a number of automated scripts to produce the nanopublications and load them into the SPARQL endpoint. An example nanopublication is shown in Fig. 7. We used the SIO (*Dumontier et al., 2014*) as a global schema to describe the nature and components of the associations, and coupled this with the PSI-MI ontology (*Hermjakob et al., 2004*) to denote the types of interactions. We used the W3C's PROV-O (*Lebo, Sahoo & McGuinness, 2013*) to capture provenance of the assertion (which data source it originated from). We loaded our nanopublications into Blazegraph, an RDF nanopublication compatible database. The data is accessed using its native SPARQL endpoint by the web application.

## Assertion probability

Each knowledge graph fragment, enclosed in a nanopublication, is assigned a probability based on the quality of the methods used to create the assertions in the fragment. We compute probabilities based on two different methods. Manually curated assertions, from DrugBank, OMIM, and COSMIC gene census, are directly given a probability $p = 0.999$. Assertions that have been derived from a specific experimental method are given probabilities appropriate for that method. These probabilities are derived from a expert-driven measure of the reliability of the experimental method used to derive the association. Factors involved in the assessment of confidence include the degree of indirection in the assay, the sensitivity and specificity of the approach, and reproducibility of results under different conditions based on the comparative analyses of techniques (*Skrabanek et al. 2008*; *Sprinzak, Sattath & Margalit, 2003*). Two expert bioinformaticians rated the reliability of each method and assigned a score of 1–3, where 1 corresponds to low confidence and 3 to high confidence. After their initial assessment, they conferred on their reasoning for each score to resolve differences where possible. The experts considered level 1 to correspond to weak evidence that needs independent verification. Level 2 methods are generally reliable, but should have additional biological evidence. Level 3 methods are high-quality method that produces few false positives. We calculated inter-annotator agreement between the two annotators over the three categories using Scott's Pi. Scott's Pi is similar to Cohen's kappa in that it improves on simple observed agreement by factoring in the extent of agreement that might be expected by chance. We determined the agreement to be 0.56 (Scott's Pi value of 0.26) across 104 experimental methods comprising of 99.9999% of interaction annotations (*Scott, 1955*).

The scores of 1, 2, and 3 were then assigned provisional probabilities of $p = 0.8$, $p = 0.95$, and $p = 0.99$ respectively. We chose these probabilities as approximations of the conceptual levels of probability for each rating by the experts, and feel that those probabilities correspond to how often an experiment at that confidence level can be expected to be

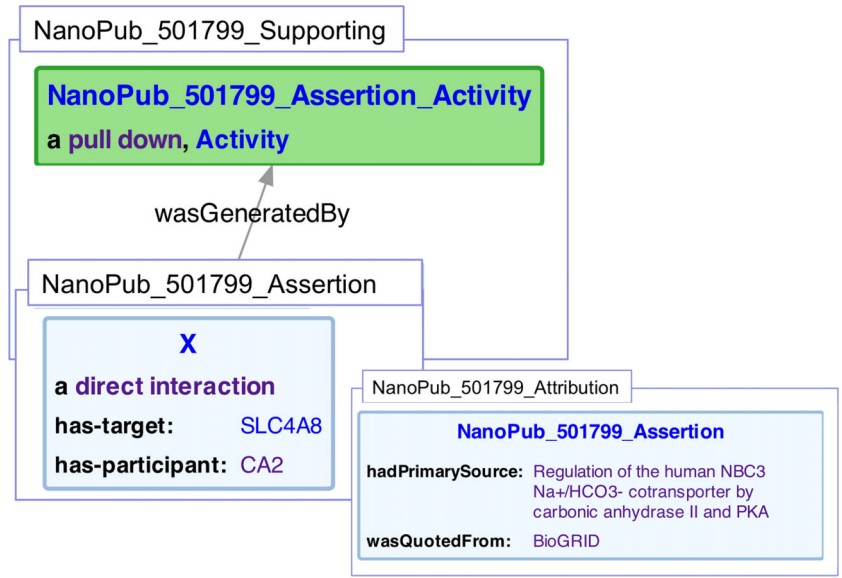

**Figure 7** **Representation of a protein/protein interaction within a nanopublication.** Three graphs are represented. The assertion graph (NanoPub_501799_Assertion), states that an interaction (X) is of type *sio:DirectInteraction*, and has the target of SLC4A8, and a participant of CA2. The supporting graph (NanoPub_501799_Supporting), states that the assertion graph was generated by a pull down experiment (one of many encoded experiment types used in, a subclass of *prov:Activity*. The attribution graph (NanoPub_501799_Attribution), in turn, states that the assertion had a primary source of (*Loiselle et al., 2004*) and that the interaction was quoted from BioGrid.

accurate. We plan to provide a more rigorous assessment of the accuracy of each method against gold standards in future work. These confidence values were encoded into an OWL ontology along with the evidence codes. The full inferences were extracted using Pellet (https://github.com/complexible/pellet) and loaded into the SPARQL endpoint, where they were used to apply the probabilities to each assertion in the knowledge graph that had experimental evidence.

## Semantic web services

We developed four SADI web services (*Wilkinson, Vandervalk & McCarthy, 2009*) in Python[1] to support easy access to the nanopublications (see Table 2) in ReDrugS. The four services are enumerated in Table 2.

   The first service is a simple free text lookup, that takes an *pml:Query*[2] (*McGuinness et al., 2007*) with a *prov:value* as a query and produces a set of entities whose labels contain the substring. This is used for interactive typeahead completion of search terms so users can look up URIs and entities without needing to know the details.

   The other three SADI services look up interactions that contain a named entity. Two of them look at the entity to find upstream and downstream connections, and the third service assumes that the entity is a biological process and finds all interactions that related to that process. The services return only one interaction for each triple (source, interaction type, target). There are often multiple probabilities per interaction, and more than one interaction per interaction type. This is because the interaction may have been recorded in multiple databases, based on different experimental methods.

[1] For further information on developing web services in Python using SADI, see this tutorial: https://github.com/markwilkinson/SADI-Semantic-Web-Services-Core/wiki/Building-Services-in-Python

[2] PML 3, in development: https://github.com/timrdf/pml. This includes PML 2 constructs that are not covered in PROV-O.

**Table 2 ReDrugS API SADI Web Services.** The API endpoint prefix is http://redrugs.tw.rpi.edu/api/.

| Service name | Description | URL | Input | Output |
|---|---|---|---|---|
| Resource text search | Look up resources using free text search against their RDFS labels. This service is optimized for typeahead user interfaces. | search | *pml:Query* | *pml:AnsweredQuery* |
| Find interactions in a biological process | Find interactions whose participants or targets also participate in the input process. | process | *sio:Process* | *sio:Process* |
| Find upstream participants | Find interactions that the input entity is a target of in and have explicit participants. | upstream | *sio:MaterialEntity* | *sio:Target* |
| Find downstream targets | Find interactions that the input entity participates in and have explicit targets. | downstream | *sio:MaterialEntity* | *sio:Agent* |

To provide a single probability score for each interaction of a source and target, the interactions are combined. A single probability is generated per identified interaction by taking the geometric mean of the probabilities for that interaction. However, this method is undesirable when combining multiple interaction records of the same type. We instead combine the interaction records using a form of probabilistic voting using composite Z-scores. This is done to model that multiple experiments that produce the same results reinforce each other, and should therefore give a higher overall probability than would be indicated by taking their mean or even by Bayes theorem. We do this by converting each probability into a Z-score (aka standard score) using the quantile function ($Q()$), summing the values, and applying the cumulative distribution function ($CDF()$) to compute the corresponding probability:

$$P(x_{1...n}) = CDF\left(\sum_{i=1}^{n} Q(P(x_i))\right)$$

These composite Z-scores, which we transform back into probabilities, are frequently used to combine multiple indicators of the same underlying phenomena, as in (*Moller et al., 1998*). However, it has a drawback. One concern is that the strategy does not account for multiple databases recording the same non-independent experiment. This can possibly inflating the probabilities of interactions described by experiments that are published in more than one database.

## Graph expansion using joint probability

In order to compute the probability that a given entity affects another, we compute the joint probability that each of the intervening interactions are true. Joint probability is the probability that every assertion in the set is true. This is computed by taking the product of probabilities of each interaction:

$$P(x_1 \wedge \ldots \wedge x_n) = \prod_{i=1}^{n} P(x_i)$$

This joint probability is used as a threshold that users can set to stop graph expansion. We also provide expansion limits using the number of interaction steps that are needed to connect the two entities.

## User interface

The user interface was developed using the above SADI web services and uses Cytoscape.js (http://cytoscape.github.io/cytoscape.js) angular.js (https://angularjs.org), and Bootstrap 3 (http://getbootstrap.com). An example network is shown in Fig. 2. Users can search for biological entities and processes, which can then be autocompleted to specific entities that are in the ReDrugS graph. Users can then add those entities and processes to the displayed graph and retrieve upstream and downstream connections and link out to more details for every entity. Cytoscape.js is used as the main rendering and network visualization tool, and provides node and edge rendering, layout, and network analysis capabilities, and has been integrated into a customized rich web client.

In order to evaluate this knowledge graph, we developed a demonstration web interface (http://redrugs.tw.rpi.edu) based on the Cytoscape.js (http://cytoscape.github.io/cytoscape.js) JavaScript library. The interface lets users enter biological entity names. As the user types, the text is resolved to a list of entities. The user finishes by selecting from the list, and submitting the search. The search returns interactions and nodes associated with the entity selected, which are added to the Cytoscape.js graph. Users are also able to select nodes and populate upstream or downstream connections. Figure 2 is an example output of this process.

## ACKNOWLEDGEMENTS

A special thanks to Pascale Gaudet who, with Michel Dumontier, evaluated the experimental methods and evidence codes listed in the protein/protein interaction ontology and gene ontology. Thank you also to Kusum Solanki and John Erickson for evaluation, feedback, and planning in the initial stages of this project.

### Funding
The authors received no funding for this work.

### Competing Interests
The authors declare that they have no competing interests.

### Author Contributions
- Jamie Patricia McCusker conceived and designed the experiments, performed the experiments, analyzed the data, contributed reagents/materials/analysis tools, wrote the paper, prepared figures and/or tables, performed the computation work, reviewed drafts of the paper.
- Michel Dumontier conceived and designed the experiments, analyzed the data, contributed reagents/materials/analysis tools, wrote the paper, performed the computation work, reviewed drafts of the paper.
- Rui Yan performed the experiments, contributed reagents/materials/analysis tools, wrote the paper, prepared figures and/or tables, performed the computation work, reviewed drafts of the paper.

- Sylvia He contributed reagents/materials/analysis tools, prepared figures and/or tables, performed the computation work, reviewed drafts of the paper.
- Jonathan S. Dordick conceived and designed the experiments, reviewed drafts of the paper.
- Deborah L. McGuinness conceived and designed the experiments, wrote the paper, reviewed drafts of the paper.

## Data Deposition

Data can be found at https://data.rpi.edu/xmlui/handle/10833/1760.

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
