# Peer review of "Finding melanoma drugs through a probabilistic knowledge graph"

_PeerJ Computer Science, doi:10.7717/peerj-cs.106_

## Round 0.1 · original submission · Major Revisions

The work itself is very interesting, but the paper can be much improved. Please extend the paper considerably by providing the details on the methodology, and a deep discussion on the results (computationally and biologically). Also provide justification on the selection of parameter on the data analytics - please refer to the reviewer's comments / requirements on this.

Reviewer 1 ·

Basic reporting

This manuscript proposes an integrated network analysis system to detect novel drug-disease associations. Various drug and disease related data sources were integrated and a probabilistic measure to evaluate the reliability of medical entity links was utilized. Several high quality melanoma drugs were generated and evaluated by recent literatures.
Due to the complicated underlying network mechanisms of disease phenotypes and the perturbation of drug treatment, it is difficult to find and confirm novel drugs for disease treatment. This manuscript do provide an interesting computing approach to find novel drug candidates for complex diseases.

Experimental design

This manuscript focuses on proposing a complex network analysis system for novel drug detection for disease treatment. The authors have introduced the related data sources, data fusion and network visualization functionalities. An instance on novel melanoma drug detection is introduced and evaluated on the reliable and novelty.

Validity of the findings

The performance of the generated results from the network analysis system has been evaluated by the measures like precision, recall and F1 measure and the results showed that the system has obtained acceptable performance. Furthermore, 25 detected melanoma drugs were validated by related literature as well as a search for registered clinical trials in ClinicalTrials.gov.

Additional comments

This manuscript describes an integrated network analysis system on drug-target-disease associations to predict novel repositioning drugs. It integrated various drug and disease related data sources, such as drug-target associations, genotype-phenotype associations and protein-protein interactions and proposed a probabilistic approach to define the direct and indirect associations between these entities. The platform consists of a graphical web application, an application programming interface (API), and a knowledge base. The knowledge graph was stored in semantic web schema and it can be visualized as interactive networks using social network packages. The system has acceptable performance on predicting novel drugs for diseases and, In particular, the platform has generated 25 high quality Melanoma drugs, which have been cross evaluated by literature query and clinicaltrial.gov database. Therefore, this is a very interesting study that has delivered meaningful and biological useful results on novel drug indication prediction. Overall, an excellent work on network pharmacology.
The minor improvement would be that a clear description on the calculation of 'joint probability' should be provided possibly in Section 4.2. Currently, there is only introduction of the principle to set the weights on direct links. However, the weight(or probability) of the indirect links(i.e. drug-disease) is a key development of the proposed system and thus should be described.

Reviewer 2 ·

Basic reporting

The structure of the paper looks reasonable. The paper is easy to follow.

Experimental design

The goal of the paper is to use an evidence-based (in the form of probability) knowledge graph to help identify novel and known drugs for melanoma. The probability is computed based on so called expert-driven measures of the reliability of experimental methods that come from existing studies. In order for other to reproduce the results, it is necessary to provide the detailed formula for computing and combining the mentioned measures, in the final draft of the article. Also, little justifications were provided to the rather high probability (0.95) for the second confidence level.

Validity of the findings

The presented knowledge graph uses 5 data sources: iRefIndex, DrugBank, UniProt Gene Ontology Annotations, the Online Mendelian Inheritance in 168 Man and the COSMIC Gene Census. Associations are represented as ontology based linked data. However, it is unclear how such associations are captured. Some important justifications are still missing, including the two thresholds of interaction steps (3) and joint probability value (0.93).

Additional comments

The paper presents a seemingly very promising approach; however, some key details are not clarified in the current draft of the paper.

---

## Round 0.2 · accepted · Accept

Very useful extension has been made on the revision. A proof-reading is needed for the final publication. Re-editing of the figures 4 and 7 is recommended to fit better with the main text.